# Fractal dimension of cortical functional connectivity networks & severity of disorders of consciousness

**Thomas F. Varley**[1,5,6]*, **Michael Craig**[1,5], **Ram Adapa**[1], **Paola Finoia**[1], **Guy Williams**[2], **Judith Allanson**[3], **John Pickard**[2,4], **David K. Menon**[1,2], **Emmanuel A. Stamatakis**[1,5]

**1** Division of Anaesthesia, School of Clinical Medicine, University of Cambridge, Cambridgeshire, England, United Kingdom, **2** Wolfson Brain Imaging Center, University of Cambridge, Cambridgeshire, England, United Kingdom, **3** Department of Neurorehabilitation, Addenbrooke's Hospital, Cambridgeshire, England, United Kingdom, **4** Division of Neurosurgery, School of Clinical Medicine, University of Cambridge, Cambridgeshire, England, United Kingdom, **5** Department of Clinical Neurosciences, School of Clinical Medicine, University of Cambridge, Cambridgeshire, England, United Kingdom, **6** Department of Psychological & Brain Sciences, Indiana University, Bloomington, Indiana, United States of America

* tvarley@iu.edu

**Data Availability Statement:** Due to patient privacy concerns, data are available upon request by qualified researchers. The UK Health Research Authority mandates that the confidentiality of data

## Abstract

Recent evidence suggests that the quantity and quality of conscious experience may be a function of the complexity of activity in the brain and that consciousness emerges in a critical zone between low and high-entropy states. We propose fractal shapes as a measure of proximity to this critical point, as fractal dimension encodes information about complexity beyond simple entropy or randomness, and fractal structures are known to emerge in systems nearing a critical point. To validate this, we tested several measures of fractal dimension on the brain activity from healthy volunteers and patients with disorders of consciousness of varying severity. We used a Compact Box Burning algorithm to compute the fractal dimension of cortical functional connectivity networks as well as computing the fractal dimension of the associated adjacency matrices using a 2D box-counting algorithm. To test whether brain activity is fractal in time as well as space, we used the Higuchi temporal fractal dimension on BOLD time-series. We found significant decreases in the fractal dimension between healthy volunteers (n = 15), patients in a minimally conscious state (n = 10), and patients in a vegetative state (n = 8), regardless of the mechanism of injury. We also found significant decreases in adjacency matrix fractal dimension and Higuchi temporal fractal dimension, which correlated with decreasing level of consciousness. These results suggest that cortical functional connectivity networks display fractal character and that this is associated with level of consciousness in a clinically relevant population, with higher fractal dimensions (i.e. more complex) networks being associated with higher levels of consciousness. This supports the hypothesis that level of consciousness and system complexity are positively associated, and is consistent with previous EEG, MEG, and fMRI studies.

is the responsibility of Chief Investigators for the initial studies (in this case, Dr. Allanson and Prof Menon; and anyone to whom this responsibility is handed – for example, in the context of retirement or transfer to another institution). For researchers interested in working with this dataset, please contact Dr. Judith Allanson (judith.allanson@addenbrookes.nhs.uk), Dr. David Menon (dkm13@cam.ac.uk) and/or Dr. Emmanuel Stamatakis (eas46@cam.ac.uk). Requests will be considered on a case-by-case basis, assessing the feasibility and appropriateness of the proposed study, and the capacity to maintain the required levels of data security, consistent with the original approved Research Ethics approval, and the patient information sheet that was the basis of consent obtained from research subjects.

**Funding:** This work was supported by grants from the Wellcome Trust Clinical Research Training Fellowship to RMA (Contract grant number: 083660/Z/07/Z); the UK Medical Research Council [U.1055.01.002.00001.01 to JDP; the James S. McDonnell Foundation to JDP; the Evelyn Trust, Cambridge to JA, the National Institute for Health Research (NIHR, UK), Cambridge Biomedical Research Centre and NIHR Senior Investigator Awards to JDP and DKM; The Canadian Institute for Advanced Research (CIFAR) to DKM and EAS; the Stephen Erskine Fellowship (Queens' College, Cambridge) to EAS; the British Oxygen Professorship of the Royal College of Anaesthetists to DKM. MC was supported by the Cambridge International Trust and the Howard Sidney Sussex Research Studentship. TFV is supported by NSF-NRT grant 1735095, Interdisciplinary Training in Complex Networks and Systems. This work was also supported by the Evelyn Trust, Cambridge and the EoE CLAHRC fellowship to J.A and the NIHR Brain Injury Healthcare Technology Co-operative based at Cambridge University Hospitals NHS Foundation Trust and University of Cambridge.

**Competing interests:** The authors have declared that no competing interests exist.

## Introduction

Research into the neural origins of conscious experience has suggested that consciousness may be associated with the "complexity" of information processing in the brain [1, 2]. While "complexity" is challenging to define, several measures have emerged in complex systems science for describing what it means for a system to be 'complex' in domains such as system architecture, spatial, and temporal dynamics [3]. In the context of biology, complexity can refer to how the components of a naturally occurring system interact and encode information, with a particular focus on emergent properties and self-organizing behaviour [4]. "complexity" is operationalized in different ways in different fields, but a common theme is to use "randomness" (often measured in terms of incompressibility) as a proxy measure for a more nebulous concept. In the study of consciousness, Lempel-Ziv compressibility ($LZ_C$) [5] is the most frequently-used measure [6–10]. In $LZ_C$ analyses a maximally random (or incompressible) signal would be indicated as having the highest "complexity." This is, however, a somewhat counterintuitive understanding of what we mean when we talk about the "complexity" of brain activity: the brain is complex not because it is highly random, but because it combines an incredible degree of order with a high degree of unpredictability. It is hard to imagine how a brain outputting algorithmically random noise could doing anything at all, let alone supporting consciousness and cognition. Not only is the brain both structured and unpredictable, it also shows one of the hallmarks of complex systems writ large: emergent dynamics over multiple scales. With this in mind, we strongly feel that consciousness science requires further discussion and refinement of what "complexity" means in the context of the brain. As a working definition, we propose that complexity should be understood as a fundamentally multi-scale phenomena, emerging in systems that display both a high degree of emergent structure and organization as well incompressible and unpredictable features.

One significant attempt to prose a more nuanced relationship between consciousness and complexity is the Entropic Brain Hypothesis (EBH). The EBH posits that consciousness emerges when the brain is near a critical zone between order and randomness, known as the critical regime, and that to move too far in either direction will result in a change in the quality of consciousness, and ultimately, loss of consciousness entirely [11, 12]. Various studies have used different metrics to approximate the complexity of brain activity, and the results have been quite consistent, even across modalities. Studies that estimate the Lempel-Ziv complexity of EEG and MEG signals have found that the algorithmic complexity of time-series is decreased in both healthy volunteers and patients who have had their level of consciousness reduced by a range of mechanisms, including sleep [8], sedation with anaesthetics [7], and brain injury [13]. Conversely, the complexity of brain signals is increased in volunteers who are under the influence of psychedelic drugs like LSD, suggesting a corresponding increase in the complexity of brain activity [9]. Studies have also found that alteration to consciousness is associated with differences in the complexity of functional connectivity networks [14–18], which may imply that the spatial complexity of brain activity is as important for the maintenance of consciousness as temporal complexity indexed by MEG and EEG measures.

A core feature of the EBH is that consciousness emerges, not where algorithmic complexity is maximal, but in the critical zone, on the boarder between low- and high-entropy states ("the edge of chaos"). Several publications suggest that the healthy brain operates at, or just below, this area of criticality [19, 20], and there are compelling theoretical reasons to prefer a critical model of the brain: in neural networks, critical systems show the greatest ability to perform computations [21], store and retrieve information [22], and criticality maximizes the range of input scales (dynamic range), due to the scale free nature of critical activity [23]. While criticality and integration of information, as formalized by Tononi and Sporns [24], are separate *in*

*vivo* and *in silico* studies have found that they are locally maximal in the same spaces [25]. While none of these results are themselves showing direct association between consciousness and criticality, the behaviours that criticality optimizes are all qualities that a brain capable of supporting a complex phenomenon like consciousness might be expected to show. It is worth noting that the specifics of criticality in the brain have not gone unchallenged [26], however, studies investigating the relationship between criticality, consciousness, and brain function in disorders of consciousness may shed further light on the topic.

One of the "fingerprints" of critical phenomena is the emergence of scale-free, or fractal, structures near the critical point [19]. As discussed above, one of the hallmarks of non-trivial complexity is the emergence of higher-scale dynamics from lower ones: scale-freeness seems 'baked-in' to any understanding of complex systems, including in the brain. There has been considerable work done already on fractal analysis of brain structures and signals [27, 28] and how changes to fractal character relate to consciousness. Research has found that the fractal dimension of brain activity is related to the level and content of consciousness: the fractal dimension of EEG signals reliably falls during sleep [29–32], sedation [33], and loss of consciousness following the administration of anaesthesia [31, 32, 34]. Temporal fractal dimension also changes during internally vs. externally generated perceptions [35] attention [36] and during altered states such as hypnosis [37]. In addition to analysis of the temporal domain, Ruiz de Miras et al., (2019) also explored the fractal dimension of the spatial distribution of significant sources following TMS-perturbation and a spatiotemporal measure of the same. Similar to the temporal measures, spatiotemporal fractal dimension also dropped when consciousness was lost under conditions of propofol anesthesia and NREM sleep. It may be that alterations to the fractal dimension of brain activity during reduced states of consciousness are reflecting movement towards or away from the critical zone [11, 12] and the information-processing capabilities it supports.

Analyses of fractal dimension of brain activity has largely been done in the temporal domain (excluding the work by Ruiz de Miras et al.), primarily using analysis of one-dimensional EEG signals. There has been far less analysis of the fractal character functional connectivity networks, which encode higher-order relationships between elements in the system [38]. Gallos et al., [39, 40] showed that there was a relationship between the fractal structure of a voxel-to-voxel level functional connectivity network and the threshold at which the edges were removed: the subgraph of strongest edges has pronounced fractal character, however, when weaker connections are incorporated, the fractal character is reduced and replaced by a small-world topology. It was further shown that these weak small-world connections provide near optimal integration of information flow between the strong fractal modules. Gallos et al. (2012), proposed that this may be a possible solution to the problem of integration versus modularity by dividing the functional connectome into two layers: the 'foundational layer,' which is made up of the strongest connections forms a large-world, fractal backbone that is modular, but not particularly well-integrated, where low-level sensory processing might occur before being bound together by higher, connected layers. The second layer, the 'integration layer' is the set of weaker edges that connect to provide integration for the different modules of the 'foundational layer' [39, 40].

Based on these results, we made two hypotheses: (1) functional connectivity networks in the neocortex would have a measurable fractal character when all of the weaker edges had been thresholded out and (2) that in patients with reduced level of consciousness, fractal dimension would be reduced. To investigate this, we used resting-state fMRI data from healthy volunteers and patients suffering from reduced levels of consciousness associated with brain injury. We divided these patients into two subgroups based on clinical diagnosis: minimally-conscious state (MCS) and vegetative state (VS), based on accepted diagnostic criteria [41]. In

general, a higher fractal dimension is associated with a more complex system [42], and so a decrease in fractal dimension in patients with reduced level of consciousness would suggest that the neural activity in those participants was reduced, which would be consistent with the predictions of the Entropic Brain Hypothesis. To supplement this data, we also used a commonly-used measure of time-series fractal dimension, the Higuchi temporal fractal dimension algorithm [43] to determine whether the fractal character of temporal activity follows the same pattern as spatial activity.

## Materials & methods

### Calculating network fractal dimension

Since the fractal dimension of most real-world systems cannot be solved analytically, researchers commonly use a family of algorithms known as box-counting measures to determine the fractal dimension of a natural system. The box-counting dimension describes how the topology of a surface changes (or remains the same) at different scales. For any shape, two values are defined: $l_B$ which is the length of an $n$-dimensional box and $N(l_B)$, which is the minimum number of boxes necessary to 'tile' the surface in question. If the shape being tiled is a fractal, then:

$$N_B(l_B) \propto l_B^{-d_B} \tag{1}$$

Where $d_B$ is the box-counting dimension. Algebraic manipulation shows that $d_B$ can be extracted by linear regression as:

$$\lim_{l_B \to 1} \frac{-ln(N_B(l_B))}{ln(l_B)} \propto d_B \tag{2}$$

A similar logic is used when calculating the box-counting dimension of a graph. For a graph $G = (V, E)$, a box with diameter $l_B$ defines a set of nodes $B \subset V$ where for every pair of nodes $v_i$ and $v_j$ the distance between them $l_{ij} < l_B$. To quantify the fractal dimension of the functional connectivity networks, a box counting method, the Compact Box Burning (CBB) algorithm [44], was used to find $N_B(l_B)$ for a range of integer $l_B$ values 1..10. If $G$ has fractal character, a plot of $ln(N_B(l_B))$ vs. $ln(l_B)$ should be roughly linear, with a slope of $-d_B$. We chose the CBB algorithm because it is an easily-implemented algorithm that can handle smaller networks than alternatives such as the maximum-excluded mass-burning algorithm [44]. We used a modified implementation of the freely-available code from https://hmakse.ccny.cuny.edu/software-and-data/.

Due to the logarithmic relationship between box-size and fractal dimension, exponentially higher resolutions (in this case, numbers of nodes) are required to achieve modest increases in the accuracy of the measured fractal dimension. Computational explorations, where a box-counting method is used to approximate a fractal dimension that has already been solved analytically, show that the box-counting dimension converges to the true dimension with excruciating slowness [45], necessitating largest network that is computationally tractable. In this context, where each node in our network maps to a specific brain region, we had to segment (parcellate) the cortex into as many distinct brain regions as we could, in this case, using a parcellation with 1000 ROIs [46].

We should note that we are not doing a truly rigorous power-law inference. The question of when an empirical distribution can be considered to follow a power-law is a rich field of research [47–49]. To do a statistically rigorous power-law inference typically requires multiple

decades of values to do a maximum likelihood estimate. Due to the inherent limitations of this box-counting algorithm, such a wide range was impossible. Consequently, we made no strong claims about how well any condition adheres to a power law, but rather, are interested in how multi-scale structure changes between conditions.

**FracLac adjacency matrix analysis.** Our second test of fractal structure used a two-dimensional box-counting method to analyse the associated adjacency matrix representations of the functional brain networks. This analysis served two purposes: primarily, it was meant to replicate the results of the CBB analysis, however we also hoped that, if it did replicate the initial results, it could be a more computationally efficient method for estimating the fractal dimension of a brain network. By using a different representation of the network, we hoped to show that the quality of network fractal dimension is conserved across isomorphic representations. This would increase our confidence in the CBB results by showing that our findings are unlikely to be an artefact of that particular algorithm. For a given graph $G = (V, E)$ with nodes $v_i$ and $v_j$, the corresponding adjacency matrix, $A(G)$ is defined:

$$A(G)_{ij} = \begin{cases} 1, & \text{if } E_{ij} \in E \\ 0, & \text{otherwise} \end{cases} \tag{3}$$

In the resulting matrix, every 1 represents an edge between two nodes $v_i$ and $v_j$. If the distribution of edges $E \in G$ is fractal, we hypothesized that the distribution of 1's in the associated matrix $A(G)$ would also have fractal character. To test this, we used the program FracLac (Karperien, A., FracLac for ImageJ, version 2015sep09)(http://imagej.nih.gov/ij/plugins/fraclac/fraclac.html), a plugin for ImageJ software (Wayne Rasband, National Institutes of Health, USA). FracLac uses a simple, 2-dimensional box-counting algorithm to return the fractal dimension of the distribution of pixels in the image. FracLac returns an upper and lower bound on the range of the fractal dimension for each image, based on the instantaneous value of $d_B$ at every value of $l_B$. For the purposes of this analysis, we took the mean of those values and defined that average as the fractal dimension of each image. The adjacency matrices were exported as binary .jpg images for analysis, and the default values for FracLac's batch image analysis were used.

We hypothesized that this method, while more accessible and less abstract than the Compact-Box-Burning algorithm, would be less sensitive to small changes in fractal dimension between conditions, as some information is lost when doing a two-dimensional box counting algorithm on a flat network representation, rather than operating directly on a graph.

## Higuchi temporal fractal dimension

We used the Higuchi temporal fractal dimension algorithm, widely used in EEG and MEG analysis, to calculate the fractal dimension of temporal brain activity [43, 50]. We will briefly describe the method here. The algorithm takes in a time-series $X(t)$ with $N$ individual samples corresponding to one Hilbert-transformed BOLD time-series extracted from our functional brain scans (details below). From each time-series $X(t)$, we create a new time-series $X(t)_k^m$, defined as follows:

$$X(t)_k^m = x_m, x_{m+k}, x_{m+2k}, \ldots, x_{m+\lfloor \frac{N-m}{k} \rfloor k} \tag{4}$$

where $m = 1, 2, \ldots, k$.

For each time-series $X(t)_k^m$ in $k_1, k_2, \ldots k_{max}$, the length of that series, $L_m(k)$, is given by:

$$L_m(k) = \frac{\left(\sum_{i=1}^{\lfloor\frac{N-m}{k}\rfloor} |x_{im+k} - x_{(i-1)k}|\right)\frac{N-1}{\lfloor\frac{N-m}{k}\rfloor k}}{k} \tag{5}$$

We then define the average length of the series $\langle L(k)\rangle$, on the interval $[k, L_m(k)]$ as:

$$\langle L(k)\rangle = \sum_{m=1}^{k} \frac{L_i(k)}{k} \tag{6}$$

If our initial time-series $X(t)$ has fractal character, then $\langle L(k)\rangle \propto k^{-D}$. As with the procedure for calculating the network fractal dimension, the algorithm iterates through values of $k$ from $1 \ldots k_{max}$ and calculates $ln(\langle L(k)\rangle)$ vs. $ln(k^{-1})$, extracting $D$ by linear regression. The various values of $k$ can be thought of as analogous to the various values of $l_B$ used to calculate the network fractal dimension. The Higuchi algorithm requires a pre-defined $k_{max}$ value as an input, along with the target time-series. This value is usually determined by sampling the results returned by different values of $k_{max}$ and selecting a value based on the range of $k_{max}$ where the fractal dimension is stable. For both DOC datasets, we chose $k_{max} = 64$ as this was the largest value that our algorithm could handle.

The implementation we used was from the PyEEG toolbox [51], downloaded from the Anaconda repository.

## fMRI data acquisition & preprocessing

**Healthy control data.** Ethical approval for these studies was obtained from the Cambridgeshire 2 Regional Ethics Committee, and all subjects gave informed, written consent to participate in the study. Twenty five healthy volunteer subjects were recruited for scanning. The acquisition procedures are described in detail in [52, 53]: MRI data were acquired on a Siemens Trio 3T scanner (Wolfson Brain Imaging Center, Cambridge). T1-weighted were acquired using an MP-RAGE sequence (TR = 2250 ms, TI = 900 ms, TE = 2.99 ms and flip angle = 9˚), with an structural images at 1 mm isotropic resolution in the sagittal plane. Each functional BOLD volume consisted of 32 interleaved, descending, oblique axial slices, 3 mm thick with interslice gap of 0.75 mm and in-plane resolution of 3 mm, field of view = 192 × 192 mm, TR = 2 s, TE = 30 ms, and flip angle 78 deg.

Of the 25 healthy volunteer datasets, 10 were excluded, either because of missing scans (n = 2), or due of excessive motion in the scanner (n = 8, 5mm maximum motion threshold). For this study, we only used the awake, control condition described in the original Stamatakis study, ignoring the drug conditions.

The resulting images were preprocessed using the CONN functional connectivity toolbox [54], which uses Statistical Parametric Mapping 12 (SPM12; http://www.fil.ion.ucl.ac.uk/spm/) and MATLAB version 2017a (http://www.mathworks.co.uk/products/matlab/). We used the default preprocessing pipeline, which includes realignment (motion estimation and correction), slice-timing correction, outlier detection, structural segmentation and normalization, de-noising with CompCor [55] and finally smoothing. A smoothing kernel of 6mm was applied, and denoising was done using a band-pass filter range of [0.008, 0.09] Hz.

**Data from patients with disorders of consciousness.** Data was acquired at Addenbrooke's Hospital in Cambridge, UK, on a 3T Tim Trio Siemens system (Erlangen Germany). Ethical approval for testing patients was provided by the National Research Ethics Service (National Health Service, UK; LREC reference 99/391). A sample DOC patients with verifiable diagnosis were recruited from specialised long-term care centers. Written consent was

obtained from the patient's legal representatives. Medication prescribed to each patient was maintained during scanning. T1-weighted images were acquired with an MP-RAGE sequence (TR = 2300ms, TE = 2.47ms, 150 slices, $1 \times 1 \times 1mm^2$ resolution). Functional images, 32 slices each, were acquired using an echo planar sequence (TR = 2000 ms, TE = 30 ms, flip angle = 78 deg, 3 x 3 x 3.75mm$^2$ resolution). Subjects were split into two groups: those who met the criteria for being in a minimally conscious state (MCS, n = 10), and those who were in a vegetative state (VS, n = 8).

Preprocessing was performed with SPM12, MATLAB, and CONN, as described above. The first five volumes were removed to eliminate saturation effects and achieve steady state magnetization. Slice-timing and movement correction (motion estimation and correction) were performed as above, including outlier detection, structural segmentation and normalization, 6 mm FWHM Gaussian smoothing, and denoising with a band-pass filter with range [0.008, 0.09] Hz. To reduce movement-related and physiological artefacts specific to DOC patients, data underwent further de-spiking with a hyperbolic tangent squashing function. Next the CompCor technique was used to remove the first 5 principal components of the signal from the white matter and cerebrospinal fluid masks, as well as 6 motion parameters and their first order temporal derivatives and a linear de-trending term [55].

## Formation of networks

After preprocessing, BOLD time-series data were extracted from each brain in CONN and the cerebral cortex was segmented into 1000 distinct ROIs, using the Schaefer Local/Global 1000 Parcellation [46] (https://github.com/ThomasYeoLab/CBIG/tree/master/stable_projects/brain_parcellation/). Due to the slow-convergence of Eq 2, and the necessity of having a network with a wide enough diameter to accommodate a sufficiently wide range of box-sizes, we attempted to strike an optimal balance between network resolution and computational tractability.

For some DOC patients, there were ROI nodes which mapped to regions that had been so damaged that no detectable signal was recovered: these time-series were removed from analysis. For the MCS patients, the average number of removed nodes was 1.2 ± 1.53 nodes ($\approx 0.12\%$ of all nodes), while for the VS patients it was 5.38 ± 7.12 nodes ($\approx 0.54\%$ of all nodes). We expect that the removal of such a comparatively small number of nodes to have a negligible effect on our overall-analysis. For each brain region, it's associated time-series $F_i(t)$ was correlated against every other time-series ($F_j(t)$), using the Pearson Correlation, forming a matrix $M$ such that:

$$M_{ij} = \rho(F_i(t), F_j(t))$$

The correlation matrix has a series of ones that run down the diagonal, corresponding the correlation between each timeseries and itself which, if treated directly as a graph adjacency matrix, would produce a graph where each node had exactly one self-loop in addition to all it's other connections. To correct for this, the matrices were filtered to remove self-loops by turning the diagonal of ones to zeros, ensuring simple graphs.

Finally, following the findings by Gallos et al. (2012), that fractal character was only present at high thresholds the matrices were binarized with a 95% threshold, such that:

$$M'_{ij} = \begin{cases} 1, & \text{if } M_{ij} \geq P_{95} \\ 0, & \text{otherwise} \end{cases}$$

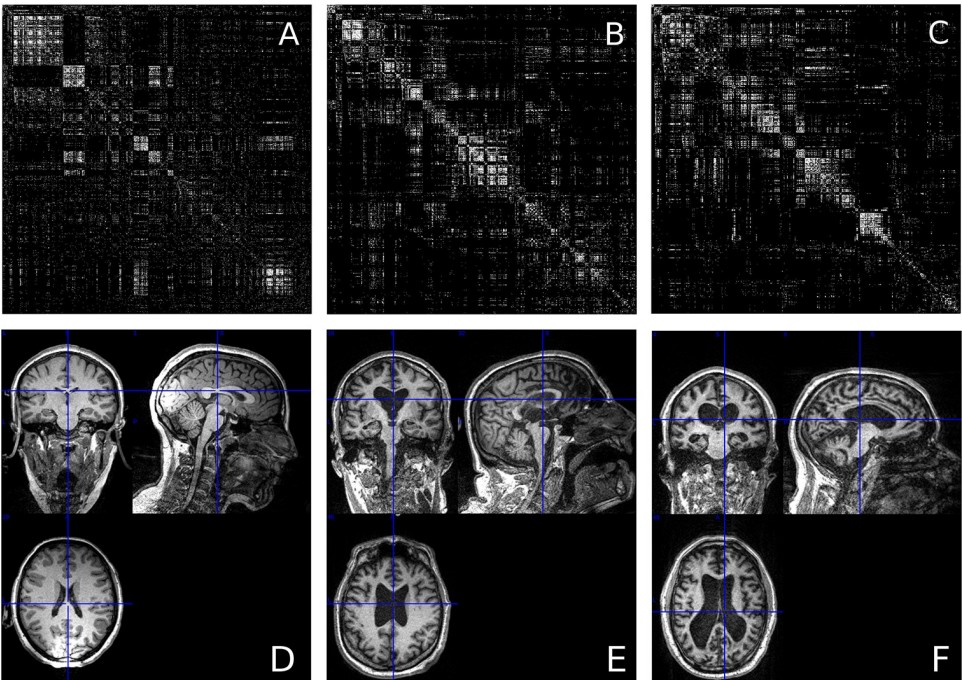

**Fig 1. Adjacency matrices by condition.** Three $1000 \times 1000$ adjacency matrices, representing the three conditions. *A* is a sample from the healthy control group, *B* is a sample from the MCS group, and *C* is a sample from the VS group, It is not immediately apparent that the fractal dimension of points in these groups is different. Below, see the associated structural scans (*D* is from a healthy control volunteer, *E* is from an MCS patient, and *F* from a VS patient). Note the increasingly cortical atrophy and expansion of ventricles as severity increases. Structural scans visualized in MRICron http://people.cas.sc.edu/rorden/mricron/index.html.

All surviving values $M_{ij} < 0 \mapsto 0$. The result is a sparse, symmetric, binary matrix, $M'_{ij}$, where all existing edges have weight 1. These matrices could then be treated as adjacency matrices defining functional connectivity graphs, where each row $M_i$ and column $M_j$ corresponds to an ROI in the initial cortical parcellation, and the connectivity between all nodes is given by Eq 3. To see samples of the binarized adjacency matrices, see Fig 1. To see a visualization of one of the networks, see Fig 2.

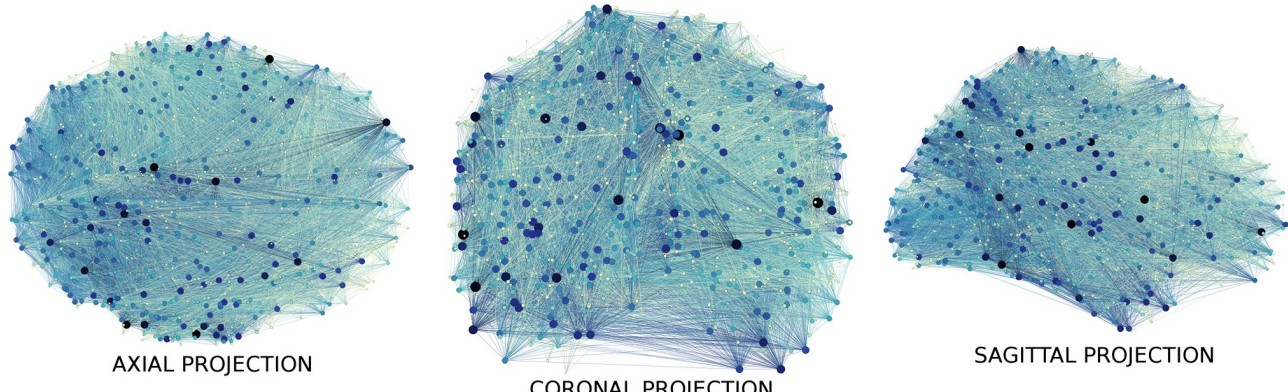

AXIAL PROJECTION          CORONAL PROJECTION          SAGITTAL PROJECTION

**Fig 2. Network visualization.** A visualization of a healthy, control functional connectivity network. Node size and darkness indicate a higher degree. Shown here are coronal, axial, and sagittal projections of the network onto a two-dimensional plane. Image made using Gephi [56] https://gephi.org/.

## Formation of null graphs

To contextualize our results in the broader space of possible graphs, we generated synthetic null networks from a variety of classes to compare our three groups of functional connectivity graphs to. Here, "null graph" refers to a network with the same number of nodes but a structure unrelated to brain connectivity. By creating these null networks, we can explore the general behaviour of the network fractal dimension algorithm and use this knowledge to inform our empirical findings. We generated three types of graph:

1. Lattices: a highly-ordered type of graph, where every node makes connections to it's $k$ nearest neighbours, where $k = 2D$ and $D$ is the embedding dimension of the graph. We tested two-dimensional and three dimensional lattices, each with 1000 nodes.

2. Random Graphs: A highly disordered type of graph, where every instance of the graph is selected at random from the space of all possible graphs. A population of 50 random graphs was generated and the average fractal dimension calculated. Each graph had 1000 nodes, and an identical number of edges to the natural functional connectivity networks, thresholded at 95%.

All null graphs were generated using the already-implemented graph generators in NetworkX [57]. We hypothesized that, despite their radically different topologies, both the lattice and random graphs would have very low fractal dimensions relative to the natural functional connectivity networks when tested with the CBB algorithm.

## Statistical analysis

All statistical analysis was carried out using Python 3.6 using the Anaconda Python environment (https://www.anaconda.com/download) and Spyder IDE (https://github.com/spyder-ide/spyder). All packages were of the newest stable release, with the exception of the NetworkX graph analysis package [57]: the implementation of the CBB algorithm required the use of NetworkX version 0.36. Given the heterogeneous nature, and small size, of the DOC datasets, a normal distribution was not assumed and all hypothesis tests were non-parametric. The analysis of variance was done using a one-way Kruskall-Wallis test, and then post-hoc testing was done using the Mann-Whitney U test. To control for false discoveries, p-values were tested with the Benjamini-Hochberg procedure with a false-discovery rate of 5% [58]. All tests were from the Scipy. Stats package [59].

## Results

### Network fractal dimension

The Kruskal-Wallis test found significant differences between the fractal dimension of functional connectivity networks for all three conditions (H(19.91), p-value $\leq$ 0.0001). The median value $d_B$ for the healthy control condition was 3.513 (IQR: 3.472-3.555), for MCS patients it was 3.309 (IQR: 3.21-3.438), and for VS patients it was 3.102 (IQR: 2.922-3.281). Post-hoc analysis with the Mann-Whitney U test found significant differences between each condition: control vs. MCS (U(13), p-value = 0.0003), control vs. VS (U(3), p-value = 0.0001), and MCS vs. VS (U(20), p-value = 0.042). For a visualization of these results see Fig 3. All p-values survived the Benjamini-Hochberg FDR correction. For a table of results see Table 1.

These results are consistent with our hypothesis that level of consciousness is positively associated with network complexity, as measured by the fractal dimension. This also shows that the direct network fractal dimension measure is sensitive enough to discriminate between

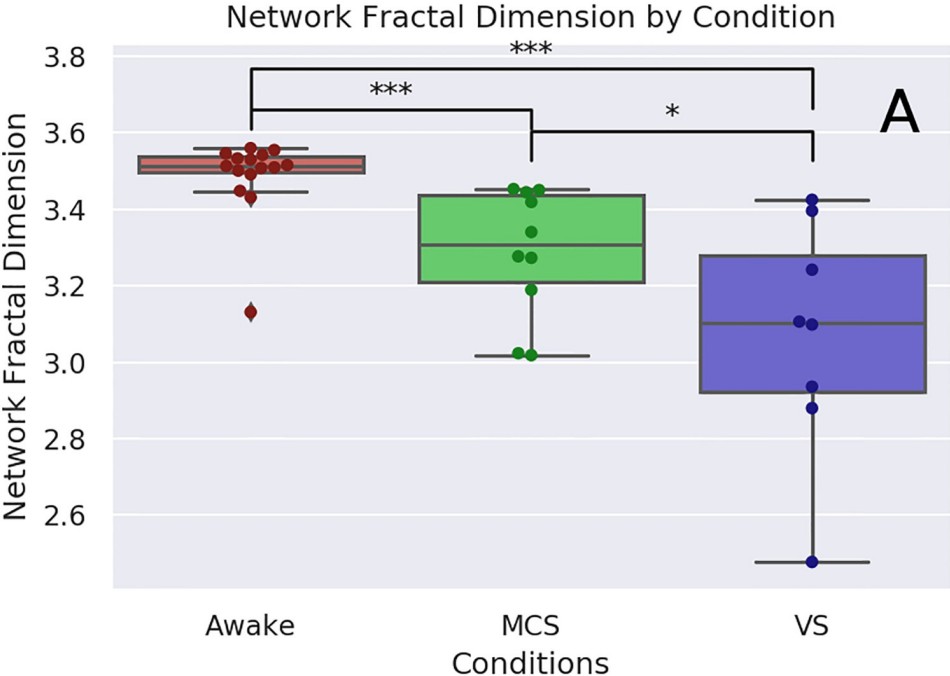

**Fig 3. Network FD analysis.** *A*: Visualization of the fractal dimension of functional connectivity networks as determined by the Compact Box Burning algorithm. Mean $d_B$ for the healthy control condition: 3.49 ± 0.1 (n = 15), for MCS patients: 3.29 ± 0.16 (n = 10), and for VS patients: 3.07 ± 0.29 (n = 8). Box length must always take integer values and does not have a regular metric unit. Post-hoc analysis with the Mann-Whitney U test found significant differences between each condition: control vs. MCS (H(41), p-value = 0.032), control vs. VS (H(23), p-value = 0.009), and MCS vs. VS (H(20), p-value = 0.042). *B* shows the relationship between $l_B$ and $N(l_B)$ in all three conditions.

**Table 1. Fractal dimension measures.**

| Metric | Healthy Control | MCS | VS |
|---|---|---|---|
| Network Fractal Dimension*** | 3.513 (IQR: 3.472-3.555) | 3.309 (IQR: 3.21-3.438) | 3.102 (IQR: 2.922-3.281) |
| Adj. Matrix Fractal Dimension** | 1.731 (IQR: 1.716-1.742) | 1.706 (IQR: 1.697-1.717) | 1.693 (IQR: 1.67-1.7) |
| Temporal Fractal Dimension*** | 1.21 (IQR: 1.2–1.23) | 0.946 (IQR: 0.927- 0.963) | 0.912 (IQR: 0.893-0.952) |

Table of results describing the how different conditions behaved under each measures of fractal dimension. Data reported are median (IQR: 25%-75%). Significance calculated using the Kruskal-Wallis analysis of variance.

** < 0.01

*** < 0.001

different clinically useful diagnoses of grey states of consciousness, rather than simply it's binary presence or absence.

## Adjacency matrix fractal dimension

The Kruskal-Wallis test found significant differences between the fractal dimensions of the adjacency matrices for the three conditions (H(10.24), p-value = 0.006). The median value for the healthy controls was 1.731 (IQR: 1.716-1.742), the median value for MCS patients was 1.706 (IQR: 1.697-1.717), and for VS patients it was 1.693 (IQR: 1.67-1.7). Post-hoc analysis with the Mann-Whitney U test found significant differences between the control and MCS conditions (U(33), p-value = 0.01), and the control and VS conditions (U(18), p-value = 0.0036), but not the VS and MCS conditions (U(25), p-value = 0.099). To ensure that our measures of network fractal dimension and adjacency matrix fractal dimension were associated, we correlated these values against each other and found a significant positive correlation (r = 0.58, p-value = 0.0005). All the significant p-values survived Benjamini-Hochberg FDR correction. For a visualization of these results, see Fig 4.

As with the direct measure of network fractal dimension, these results show that complexity is associated with level of consciousness. While this method is sensitive enough to differentiate between healthy controls and patients with disorders of consciousness, unlike the direct measure, it was not able to discriminate between disorders of consciousness of varying severity.

## Higuchi temporal fractal dimension

Kruskal-Wallis test found significant differences between all three conditions (H(25.1), p-value = $3.5 \times 10^{-6}$). The median value for the Awake patients was 1.21 (IQR: 1.2—1.23). The median value for the MCS patients was 0.946 (IQR: 0.927- 0.963), and for VS patients it was 0.912 (IQR: 0.893-0.952). Testing with the Mann-Whitney U test found a significant difference between the Awake and MCS conditions (U(0), p-value = $1.8 \times 10^{-5}$), the Awake and VS conditions (U(0), p-value = $6.13 \times 10^{-5}$) and the MCS and VS conditions (U(17), p-value = 0.023). For visualization of these results, see Fig 5. We should note that these timeseries were not all the same length: the Awake condition included 150 TRs, while the DOC scans had 300 TRs. Consequently, the comparisons between the Awake and DOC conditions should be treated with caution. To determine if there was an effect of size, we re-ran the analysis after truncated the DOC scans to the same length as the Awake scan. Analysis of variance found no significant differences between the three truncated conditions (H(4.5), p-value = 0.1). There was a significant difference between the Awake and VS conditions (H(28), p-value = 0.02.). The overall pattern was conserved: the median value for the Awake condition was the highest (1.214, IQR: 1.2-1.23), followed closely by the MCS condition (1.211, IQR: 1.18-1.24), with the VS condition coming in last (1.17 IQR: 1.12-1.23). It is surprising that the shorter BOLD timeseries had

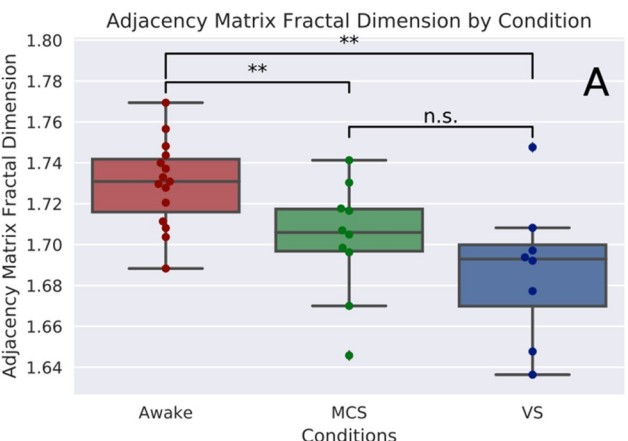

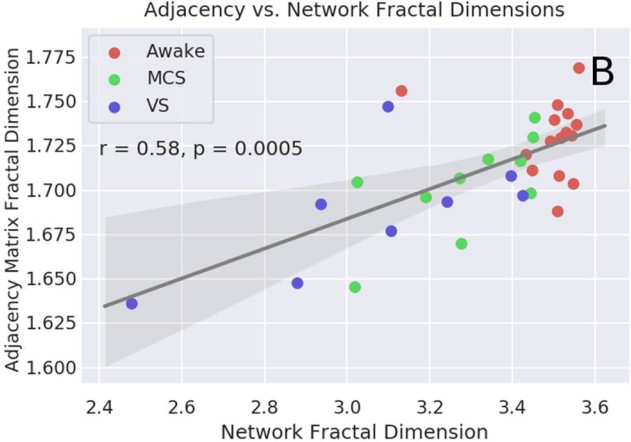

**Fig 4. Adjacency matrix FD analysis.** *A*: Visualization of the fractal dimension of functional connectivity networks as determined by FracLac analysis of the isomorphic two-dimensional adjacency matrix. Median value for the healthy controls: 1.731 (IQR: 1.716-1.742), for the MCS patients: 1.706 (IQR: 1.697-1.717), and for VS patients: 1.693 (IQR: 1.67-1.7). Post-hoc analysis with the Mann-Whitney U test found significant differences between the control and MCS conditions (H(33), p-value = 0.01), and the control and VS conditions (H(18), p-value = 0.0036), but not the VS and MCS conditions (H(25), p-value = 0.099). *B*: shows the correlation between the network fractal dimension (as calculated with the compact box burning algorithm), and the associated adjacency matrix fractal dimension (as calculated with FracLac).

a higher HFD than the longer version, the significance of this is unclear, however, the persistence of the overall pattern (Awake > MCS > VS) suggests the result is robust.

While preliminary, these results are nicely consistent with our initial hypothesis, that level of consciousness is positively associated with the fractal dimension of brain activity. Furthermore, these results complement the findings from the network fractal dimension by showing that the fractal dimension of brain activity's relationship to consciousness is measurable in temporal, as well as spatial, dimensions.

## Null graph network fractal dimension

As predicted, all the classes of null graphs had much lower fractal dimensions than any of our brain networks, as calculated by the CBB algorithm. The 2-dimensional lattice graph with 1000 nodes had a fractal dimension of $\approx 0.15$. The 3-dimensional lattice with the same number of nodes had a fractal dimension of $\approx 0.184$. The set of random networks had a higher fractal

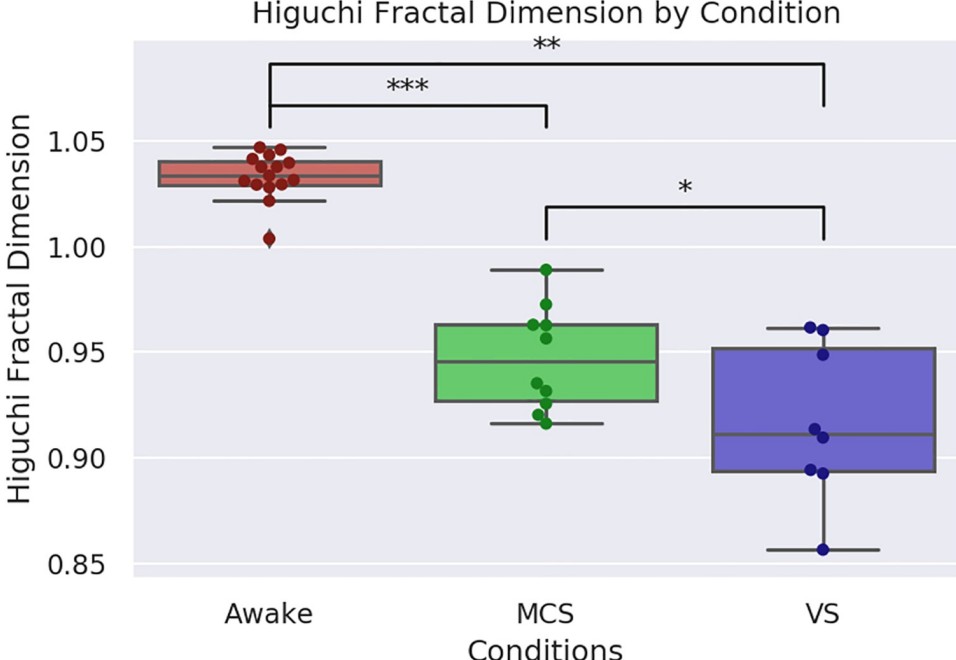

**Fig 5. Visualization of Higuchi FD analysis.** Visualization of the difference in Higuchi temporal fractal dimension between the Awake, MCS and VS conditions. The Awake condition had the highest FD with a median value of 1.21 (IQR: 1.2–1.23). As expected, the MCS condition had the next highest dimension, with a median value 0.946 (IQR: 0.927-0.963) followed by the VS condition, with a median of 0.912 (IQR: 0.894-0.952). The Wilcoxon signed-rank test found a significant difference between the MCS and VS conditions (U(17), p-value = 0.023) and between both conditions and the Awake condition (U(0), p-value = $1.8 \times 10^{-5}$, MCS) and (U(0), p-value = $6.13 \times 10^{-5}$).

dimension, although it was still far lower than any of the real functional connectivity networks, with a median value of 0.279 (IQR: 0.279, 0.2792).

These results show that the fractal dimension measure is distinct from a measure of order/randomness, as both highly ordered networks and highly random networks return similarly low values as compared to the functional connectivity networks.

## Discussion

In this study we found that the complexity of functional connectivity networks, as measured by the fractal dimension, was significantly associated with level of consciousness in healthy volunteers and patients with DOC of varying severity. When calculated using the Compact Box-Burning (CBB) Algorithm [44], the fractal dimension of these networks differentiates between healthy volunteers, patients in MCS, and patients in VS. A related box-counting algorithm, when applied to a two dimensional matrix isomorphic to the original graphs returned a similar result, although with less discriminative power.

Emergent scale-freeness is one of the hallmarks of critical dynamics [19] and so these results appear to fit with the previously described Entropic Brain Hypothesis [12], which predicts that as the brain moves further from the zone of criticality, level of consciousness falls. If the fractal character is indicative of critical behaviour, then these results may show an association between decreased signs of criticality and disorders of consciousness and a reduction in computational and information processing capabilities in the nervous system. This, in turn may explain the decrease in cognitive and behavioural complexity that are the hallmarks of disorders of consciousness. While fractal dimension and entropy are distinct concepts, entropy,

in computational models, positively correlates with fractal dimension [60, 61]. The benefit of the fractal dimension measure, however, is that it goes beyond the order/randomness binary indexed by entropic measures such as Lempel-Ziv complexity or Shannon entropy [10, 62].

The results presented here are consistent with previous work of ours using similar techniques on fMRI data from adult volunteers under the influence of LSD and psilocybin [63]. There we found that both LSD and psilocybin significantly increased the fractal dimension of high-threshold FC networks, and LSD increased the Higuchi fractal dimension of BOLD signals. In the current study, a decrease in temporal and spatial fractal dimension is associated with loss of consciousness, while in the previous case, an increase in fractal dimension is associated with a subjective increase in the "complexity" or vividness of conscious experience. Taken together, these complementary results provide converging lines of evidence that temporal and spatial fractal dimension are meaningfully related to a range of conscious states and that such analyses is feasible using BOLD signals.

One difficulty of much of this work is creating an intuitive understanding of what it means for a system to have a "lower" fractal dimension than another. Considerable previous research has shown that loss of consciousness is associated with lower fractal dimensions [29–34], but what does that mean? Different fractal shapes may have higher or lower fractal dimensions, but that doesn't necessarily mean that they are "more" or "less" fractal. In the context of a box-counting analysis, where $d_B \propto -ln(N_B(l_B))/ln(l_B)$, a higher fractal dimension corresponds to a steeper slope: small changes in $l_B$ correspond to comparatively more dramatic changes in $N(l_B)$. This may indicate a "rougher" topology, with a more heterogenous distribution of high- and low-density regions. Ruiz de Mira et al., (2019) discuss fractal dimension in terms of both integration and differentiation, suggesting that alterations to the fractal dimension may represent differences in the ability of a system to balance these competing properties.

To discuss the aetiology of the changes in network fractal dimension, we turn to previous studies, which have shown that the cerebral cortex has fractal characteristics and that changes to the fractal dimension of both the grey matter and white matter are associated with changes in cognition and the presence of clinically relevant conditions [64–67]. We hypothesize that the damage to the physical cortex by brain injury translates into changes in the fractal dimension of micro-scale structural characteristics of the cortex and that this alters how individual brain regions are able to communicate. We propose that a future study that uses this same dataset to quantify changes in the fractal dimension of physical characteristics of these brains may lend evidence to this hypothesis. Specific areas of inquiry are the fractal dimension of the folds in the neocortex, which have been previously characterized as fractal, and the network of white-matter tracts revealed by DTI imaging. It would be very interesting to perform the same analysis we have reported here on a network of white-matter connections, so long as the resolution of the resulting network is high enough to support the CBB algorithm.

There are several limitations for this study that are worth considering and suggest a need for further validation. We acknowledge the comparatively small sample size, particularly in the VS condition. As previously mentioned, the requirements of fMRI image processing demand images of brains from individuals with reduced levels of consciousness, but are not so geometrically distorted as to make registration into MNI space impossible. This puts a limit on the number of brains eligible for inclusion in this kind of study. There is also the issue of parcellation resolution: we tried several different parcellations of various sizes, but only the parcellation with 1000 ROIs had a high enough resolution to return a meaningful result, and even that was still too small to permit more than 10 integer values for $l_B$. There is also a question about the Higuchi fractal dimension results: HFD typically ranges between 1-2 and so is abnormally low in both the MCS and VS conditions. This result is reasonably robust to the $k$ parameter

chosen (as described in the methods section), and we hypothesize that it may be a result of using a time-series that is both comparatively short, and consisting of very low-frequencies. Finally, while these results have been discussed in the context of the critical brain hypothesis, we are careful to note that the question of how to infer a critical state from empirical data is a non-trivial one (for review see Timme et al., 2016) and the emergence of power-laws is only one of many tests that must be run before one can confidently make an attribution of "true" criticality. We are planing future studies more explicitly exploring the relationship between temporal and spatial fractal dimension and other measures of criticality in reduce and altered states of consciousness.

These results point to several possible subsequent investigations. First, how do temporal and spatial fractal dimensions relate to other temporal (e.g. Lempel-Ziv complexity, sample entropy, etc) and network domains (scale-freeness, small-worldness, etc)? Previous studies have used power-law, or heavy-tailed degree distributions as an indicator of fractal structure [68] and associated changes to the scaling exponent with alterations to consciousness [69]. While the degree distribution and box-counting describe different aspects of the network structure [70], the relationships between them seem like a fruitful area to explore. Of particular interest is how different network topologies facilitate the propagation of information, which may give insight into how changes to brain structure and connectivity alter information processing and integration. Similarly, previous work on discriminating states consciousness has indicated that multiple measures working in concert work better than any single unidimensional scaler [32, 71], as different measures describe different aspects of the system. Interrogating how different temporal and spatial measures interact may provide insight both into the nature of consciousness and the behaviour of these formalisms.

Going forward, we hope that this kind of analysis may one day be useful in a clinical context for estimating whether consciousness is present in patients who may be unable to give a voluntary behavioural affirmation of awareness. The fractal dimension measure encodes significant information about the complexity of a system into a single, easily digestible measure that seems to be relevant in a clinically meaningful population. As, at least in larger hospitals, MRI scans are already a routine part of clinical assessments in cases of brain damage, this measure could be incorporated into the normal course of treatment.

## Conclusion

In this study, we show that high-resolution, cortical functional connectivity networks have fractal characteristics and that, in patients with disorders of consciousness induced by traumatic brain injury or anoxic brain injury, reduction in the fractal dimension is associated with more severe disorders of consciousness. This is consistent with theories that associate the content, and quality, of consciousness with the complexity of activity in the brain. Furthermore, we believe that, with refinement, this measure may inform diagnosis and stratification in a clinical setting where physicians need to make judgements about a patients consciousness in the absence of behaviourally unambiguous indicators.

## Supporting information

**S1 Results. Disorders of consciousness results.** These are the results for the network fractal dimension, adjacency matrix fractal dimension, and higuchi fractal dimension for each of the three conditions (Awake, MCS, VS).
(CSV)

**S2 Results. Random graph results.** These are the results for the network fractal dimension analysis of the lattice networks and the ER random networks.
(CSV)

## Acknowledgments

This work was supported by grants from the Wellcome Trust Clinical Research Training Fellowship to RMA (Contract grant number: 083660/Z/07/Z); the UK Medical Research Council [U.1055.01.002.00001.01 to JDP; the James S. McDonnell Foundation to JDP; the Evelyn Trust, Cambridge to JA, the National Institute for Health Research (NIHR, UK), Cambridge Biomedical Research Centre and NIHR Senior Investigator Awards to JDP and DKM; The Canadian Institute for Advanced Research (CIFAR) to DKM and EAS; the Stephen Erskine Fellowship (Queens' College, Cambridge) to EAS; the British Oxygen Professorship of the Royal College of Anaesthetists to DKM. MC was supported by the Cambridge International Trust and the Howard Sidney Sussex Research Studentship. TFV is supported by NSF-NRT grant 1735095, Interdisciplinary Training in Complex Networks and Systems. The Evelyn Trust, Cambridge and the EoE CLAHRC fellowship to J.A; this research was also supported by the NIHR Brain Injury Healthcare Technology Co-operative based at Cambridge University Hospitals NHS Foundation Trust and University of Cambridge. We would like to thank Victoria Lupson and the staff in the Wolfson Brain Imaging Centre (WBIC) at Addenbrooke's Hospital for their assistance in scanning. We would like to thank Dian Lu and Andrea Luppi for useful discussions, and all the participants for their contribution to this study.

## Author Contributions

**Conceptualization:** Thomas F. Varley, Emmanuel A. Stamatakis.

**Data curation:** Michael Craig, Ram Adapa, Paola Finoia, Guy Williams, Judith Allanson, John Pickard, David K. Menon, Emmanuel A. Stamatakis.

**Formal analysis:** Thomas F. Varley.

**Funding acquisition:** Paola Finoia, Guy Williams, Judith Allanson, John Pickard, David K. Menon, Emmanuel A. Stamatakis.

**Investigation:** Thomas F. Varley.

**Methodology:** Thomas F. Varley, David K. Menon, Emmanuel A. Stamatakis.

**Project administration:** Thomas F. Varley, David K. Menon.

**Software:** Thomas F. Varley, Michael Craig.

**Supervision:** Michael Craig, David K. Menon, Emmanuel A. Stamatakis.

**Writing – original draft:** Thomas F. Varley.

**Writing – review & editing:** Michael Craig, David K. Menon, Emmanuel A. Stamatakis.

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
