## [Decision Letter · Decision Letter 0]

24 Oct 2019

PONE-D-19-27002

Fractal Dimension of Cortical Functional Connectivity Networks Predicts Severity in Disorders of Consciousness

PLOS ONE

Dear Mr Varley,

Thank you for submitting your manuscript to PLOS ONE. After careful consideration, we feel that it has merit but does not fully meet PLOS ONE’s publication criteria as it currently stands. Therefore, we invite you to submit a revised version of the manuscript that addresses the points raised during the review process.

According to referees' suggestions (see detailed comments below), some of the conceptual and methodological approaches here carried out have to be clarified. It is important here to introduce the meaning of the developed concepts, in order to avoid misconceptions and misinterpretations of the obtained results. I agree with referee#3 that the word "prediction" should be replaced in the title by a more appropriate one.

We would appreciate receiving your revised manuscript by Dec 08 2019 11:59PM. To enhance the reproducibility of your results, we recommend that if applicable you deposit your laboratory protocols in protocols.io, where a protocol can be assigned its own identifier (DOI) such that it can be cited independently in the future. For instructions see: http://journals.plos.org/plosone/s/submission-guidelines#loc-laboratory-protocols

We look forward to receiving your revised manuscript.

Kind regards,

Francisco J. Esteban, Ph.D., M.Sc.

Academic Editor

PLOS ONE

Journal Requirements:

3. Please provide additional details regarding participant consent. In the Methods section, please ensure that you have specified what type of consent you obtained (for instance, written or verbal) and whether the ethics committee approved this consent procedure. If verbal consent was obtained please state why it was not possible to obtain written consent and how verbal consent was recorded. If your study included minors, state whether you obtained consent from parents or guardians.

4. We understand that the manuscript may use similar methods as the following preprint:

https://www.biorxiv.org/content/10.1101/517847v2

We would be grateful if you could cite this work (either in preprint or published form) in your revised manuscript and discuss how the two submissions relate to one another. In the Methods, if you feel it necessary to repeat published protocol details, please rephrase if possible, indicate clearly that you are reproducing information posted elsewhere (e.g. “As described in detail previously [ref],…), and cite the relevant sources.

Reviewers' comments:

Reviewer's Responses to Questions

**Comments to the Author**

1. Is the manuscript technically sound, and do the data support the conclusions?

Reviewer #1: Yes

Reviewer #2: Yes

Reviewer #3: Partly

2. Has the statistical analysis been performed appropriately and rigorously? 

Reviewer #1: Yes

Reviewer #2: Yes

Reviewer #3: Yes

3. Have the authors made all data underlying the findings in their manuscript fully available?

Reviewer #1: No

Reviewer #2: Yes

Reviewer #3: Yes

4. Is the manuscript presented in an intelligible fashion and written in standard English?

Reviewer #1: Yes

Reviewer #2: Yes

Reviewer #3: Yes

5. Review Comments to the Author

Reviewer #1: It is not mathematically elegant the way in which the matrix M is defined on p. 11. The value of M_{ij} is defined in terms of itself. It makes sense in a programming setting where M is a variable whose value changes. If the authors want to explain successive filters that were applied to M, they can introduce M’, M’’ and then M, or something like that. Finally, “All surviving values M_{ij} < 0 ...”. Is it correct? If I understood the procedure, at that moment all values were 0 or 1.

With respect to the use of “null graphs” or “null networks”, it can be ambiguous at it can refer to graphs with no edges. Please indicate that you refer to a “null model” or graphs not representing brain connectivity.

Some typos:

- There is a missing reference on p. 3 “including sleep (?)”

- Unbalanced parenthesis on Eq. (6)

- In the 7th line of p. 10 there is a quotation mark that can be removed: during scanning.” T1-weighted images

- Also at the bottom of p. 10, the quotation marks in ”Schaefer Local/Global 1000 Parcellation” can be removed.

- In the caption of Fig. 1, I think “volunteers” may be changed to “volunteer”

- p. 15: “Due to the large different in scan-lengths between”. Difference?

Reviewer #2: This paper presents a study on the analysis of FD of cortical functional networks comparing three groups: healthy controls, MCS patients and VS patients. Three different FD analyses were performed: HFD, box-counting and CBB. Significant differences in FD were found between groups, with FD decreasing as the level of consciousness decreases.

The paper is clear and well written and it was a pleasure for me to read it. Conclusions are based on the results obtained. The topic of the paper has high interest.

Nevertheless, I have some comments and suggestions for the authors:

1. More relevant and related studies should be cited and compared in the paper regarding fractal analysis applied to consciousness, and fractal analysis of brain networks. For example, in the second paragraph of page 4, references Ieva 2014 and Ieva 2015 are very generic. HFD studies on consciousness at the end of Introduction (page 5) are relative old. Please, cite, comment and compare current studies on FD and consciousness such as Ruiz de Miras et al. 2019 and other references in that study: “Fractal dimension analysis of states of consciousness and unconsciousness using transcranial magnetic stimulation”. Computer Methods and Programs in Biomedicine 175, pp. 129-137. 2019.

2. Page 6. “To quantify the fractal dimension …, the Compact Box Burning (CBB) algorithm was used…”. Please, explain the reasons why that method for computing the FD of networks was selected, and include an appropriate reference to the paper describing the method. Please, clarify whether the author used any third-party software for computing the CBB or a home-made program.

3. Page 11. Definitions of F(t) and Hi(t) are missing.

4. Page 14. Table 1 would provide more information if it includes which comparisons have a p-value < 0.05

5. Page 13. Statistical analysis. It is a bit strange for me that authors have used Python for statistical analysis instead of using SPSS or MATLAB as usual in the field. Please, explain.

6. Page 15. 3.3. Higuchi temporal fractal dimension. Authors claim that they did not include the HFD analysis on awake subjects because of the different and smaller number of samples regarding DOC subjects. However, this reviewer thinks that this analysis should be included, explaining the possible limiting factors.

7. Page 15. “Surprisingly, we found no significant correlation between temporal fractal dimension and network fractal dimension …”. It is not such a surprise, since related studies, such as Ruiz de Miras et al. 2019, are in the same line. HFD needs be complemented with other measures to adequately characterize the signal.

Other minor comments are:

- Abstract, page 5, 8, 9 and 10. Even though BOLD is a common term in the field, it would be good to provide the formal noun at the first occurrence.

- Page 3: “… who have had their level of consciousness reduced by a range of mechanisms, including sleep (?), sedation …”. Missing reference.

- Page 6: “… the surface in question. the shape …”. Revise punctuation.

- Page 11: “… following the findings by Gallos et al., that …”. Provide complete reference.

- Page 11. “All surviving values Mij …”. Revise punctuation.

- Page 12. “All null graphs … in NetworkX”. Provide reference.

- Page 16. “… the fractal dimension of a these networks …”. Revise the sentence.

- Page 16. “… in the nervous system, which is turn may explain …”. Revise the sentence.

Reviewer #3: In this research report entitled “Fractal dimension of cortical functional connectivity networks predicts severity in disorders of consciousness” the authors explored the FD of functional connectivity networks and time series of BOLD signals from healthy participants and patients with disorders of consciousness (MCS, VS). They found that in all measures the FD for healthy participants was higher than for patients; and it was higher for MCS than VS patients. These results might indicate that the FD of functional structures of the brain is sensitive to abnormal states of consciousness.

General comment

The structure of the paper is convenient and it is clearly developed along the manuscript. The theoretical background is also clear and simple to explain but in my opinion some confusion exists on the use of terms as fractal dimension, fractal character and complexity. I will raise this issue in the next paragraph section.

The study is well described and its predictions are straightforward. However, I do not think it is the best experimental design. Although the results are very easy to interpret, I believe additional analyses would provide more information about the nature of the differences between the FD in the experimental groups. I will comment this point in the section about the results.

Comments on theoretical concepts

1- One of the more repeated concepts in the manuscript is ‘complexity’. I believe that in general we need to take care when use it because in most studies it is a synonym of ‘randomness’. In this study complexity is identified with a system between order and randomness, with being more or less fractal or with the level of FD that a given signal exhibits. From my point of view, complexity is not a well defined concept in science. Even in the so called science of complexity there is no agreement about the exact meaning of it. Hence, the authors need to define what complexity is in the framework of this particular investigation. I encourage the authors to give a specific working definition for complexity in the introduction section and follow it in the rest of the manuscript.

2- Another confusing point, at least for me, is that in this work it is suggested that one of the goals is to study if the functional networks are fractal-like (goal 1 in page 5). One might investigate this particular question by looking the small world properties or power law characteristics, etc.; but I believe that the FD alone does not indicate if a given object or network is fractal. It is suggested here that if the FD is higher for a given network it would show more fractal properties than other network that exhibits less FD. This is not necessarily true. The definition of a fractal is any structure made of copies of itself (self- similar) and depending of the nature of its self-similarity the FD will be high or low. For example, The Koch curve has a FD of 1.26, the Sierpinski triangle has a FD of 1.58, and I would not say that the latter is more fractal than the former. We might also use an empirical example: it is very likely to obtain a very high Higuchi’s fractal dimension for a random signal, and it does not mean that this is self affine or fractal. In general, FD indicates the density of the structure being measured, or the roughness of the object. It would be possible that in a given context a high FD indicate that it is a fractal and I believe that this is what it needs to be clarified in the manuscript. It would be needed to justify why a higher FD for a functional network means that it is a fractal. I know that in theory any topological dimension that is fractional belongs to a fractal structure but this is just for mathematical objects. Natural objects can show fractional dimensions calculated with algorithms of approximation without being fractal-like. For example, if you use the Higuchi’s fractal dimension with a simple sinusoidal curve, the FD would be slightly higher than 1. And this signal is not a fractal indeed.

3- I believe that the relationship between consciousness and complexity is not very clear in the manuscript. Complexity is related in this work with criticality as well as with the concept of complexity developed in Tononi’s theory. This is quite confusing and difficult to understand by a reader who is not familiar with these two perspectives. I would define the theoretical framework with more precision: A) Is this study testing differentiation and integration as a marker of consciousness? In this case, complexity is very well defined and one might discuss the results in this context. B) Is this study directed to find fractal structures and relate them with conscious states? Then I would try to link complexity with fractal dimension, as I suggested above, and give a solid argument to conceive FD as an indicator of self-affinity.

Comments on the methodology and the FD measures used in the study

1. A specific question I would like to raise here is Do authors really address the goal one of the study introduced in page 5? In order to state that networks constructed suppressing weak edges, have a fractal character, it would be convenient to obtain a different type of functional network and show that it is not fractal (or it is less fractal). Authors could have obtained networks with weak edges and use them for comparison. Moreover, this design would be also convenient to show if the FD decreases only in the networks with fractal characteristics vs networks with less fractal structure.

2. In this section I also want to point out that I am a bit confused about the values of the measures obtained in the study. There are three different measures. The first one is the FD of the Networks (NFD); the second one is the FD of the Adjacency matrix (AdFD); and the third one is the Higuchi’s FD (HFD).

In principle one might expect that a network maximum value of NFD would be 3 just because the maximal density of an object in a three dimensional space is 3. However, obtained values are higher than 3. Because I am not an expert in the box counting algorithm presented here, I might be wrong in this point.

The values of the AdFD measures are between one and two and this is reasonable because one might expect that FD=2 will be the more compact object in this space. But the values of HFD are a bit weird because they should range between 1 (straight line) and 2 (a very dense signal filling the entire 2 dimensional space). The means reported here are below 1. I believe it might be due to the selection of the k parameter but the authors need to check if there is an error in the estimation.

3. It would be useful to include the characteristics of the BOLD signals used in the study. At least authors should report the length of the segments, and the AD rate at which they were registered. Does ‘samples’ in section 3.3 refer to number of points in a segment? Is there only one segment per participant?

Comments on the results

The non parametric statistical approach introduced by the authors is adequate. The results showed that the FD was different for each group and measure. However, I would not say that one might predict (as suggested in the title of the manuscript) the severity of the disorder of consciousness. I think that in order to predict it would be necessary to include a larger sample of participants as well as a mathematical model of prediction (ROC curves, linear or non linear multiple regression, or a more sophisticated classifier). Hence I would say that the discussion should consider this fact and a different verb should be used in the title.

6. PLOS authors have the option to publish the peer review history of their article (what does this mean?). If published, this will include your full peer review and any attached files.

Reviewer #1: No

Reviewer #2: No

Reviewer #3: Yes: Antonio J. Ibáñez-Molina

---

## [Author Response · Author response to Decision Letter 0]

24 Nov 2019

We thank the editorial team for their input and the reviewers for their response. Please find the point-by-point responses attached as a PDF. Thank you.

---

## [Decision Letter · Decision Letter 1]

18 Dec 2019

Fractal Dimension of Cortical Functional Connectivity Networks & Severity of Disorders of Consciousness

PONE-D-19-27002R1

Dear Dr. Varley,

We are pleased to inform you that your manuscript has been judged scientifically suitable for publication and will be formally accepted for publication once it complies with all outstanding technical requirements.

With kind regards,

Francisco J. Esteban, Ph.D., M.Sc.

Academic Editor

PLOS ONE

Additional Editor Comments (optional):

Reviewers' comments:

Reviewer's Responses to Questions

**Comments to the Author**

1. If the authors have adequately addressed your comments raised in a previous round of review and you feel that this manuscript is now acceptable for publication, you may indicate that here to bypass the “Comments to the Author” section, enter your conflict of interest statement in the “Confidential to Editor” section, and submit your "Accept" recommendation.

Reviewer #1: All comments have been addressed

Reviewer #2: All comments have been addressed

Reviewer #3: All comments have been addressed

2. Is the manuscript technically sound, and do the data support the conclusions?

Reviewer #1: Yes

Reviewer #2: Yes

Reviewer #3: Partly

3. Has the statistical analysis been performed appropriately and rigorously? 

Reviewer #1: Yes

Reviewer #2: Yes

Reviewer #3: Yes

4. Have the authors made all data underlying the findings in their manuscript fully available?

Reviewer #1: No

Reviewer #2: (No Response)

Reviewer #3: Yes

5. Is the manuscript presented in an intelligible fashion and written in standard English?

Reviewer #1: Yes

Reviewer #2: Yes

Reviewer #3: Yes

6. Review Comments to the Author

Reviewer #1: All my comments have been addressed by the authors. I consider that the paper can be published in its current form.

Reviewer #2: (No Response)

Reviewer #3: I have read the review sent by the authors and I would like to indicate that all potential issues I commented were adequately addressed. Terms as fractality, complexity or fractal dimension are clearer now throughout the manuscript.

I see that some technical aspects have been improved as well.

I appreciate the chage in the title, I think it is more precise now

Because I do not have further questions I recommend this work for publication

7. PLOS authors have the option to publish the peer review history of their article (what does this mean?). If published, this will include your full peer review and any attached files.

Reviewer #1: No

Reviewer #2: No

Reviewer #3: Yes: Antonio J. Ibáñez-Molina

---

## [Editor Report · Acceptance letter]

22 Jan 2020

PONE-D-19-27002R1 

Fractal Dimension of Cortical Functional Connectivity Networks & Severity of Disorders of Consciousness 

Dear Dr. Varley:

I am pleased to inform you that your manuscript has been deemed suitable for publication in PLOS ONE. Congratulations! Your manuscript is now with our production department. 

With kind regards,

on behalf of

Dr. Francisco J. Esteban 

Academic Editor

PLOS ONE